# A Dynamic Active Safe Semi-Supervised Learning Framework for Fault Identification in Labeled Expensive Chemical Processes

**Xuqing Jia [1], Wende Tian [1,*], Chuankun Li [2], Xia Yang [1], Zhongjun Luo [3] and Hui Wang [3]**

[1] College of Chemical Engineering, Qingdao University of Science & Technology, Qingdao 266042, China; jiaxq1994@163.com (X.J.); yangxia@qust.edu.cn (X.Y.)

[2] State Key Laboratory of Safety and Control for Chemicals, SINOPEC Qingdao Research Institute of Safety Engineering, Qingdao 266071, China; lichk@163.com

[3] Shandong Qiwangda Group Petrochemical CO., LTD, Linzi 255400, China; luozj003@126.com (Z.L.); qwdwh@sina.com (H.W.)

\* Correspondence: tianwd@qust.edu.cn

**Abstract:** A novel active semi-supervised learning framework using unlabeled data is proposed for fault identification in labeled expensive chemical processes. A principal component analysis (PCA) feature selection strategy is first given to calculate the weight of the variables. Secondly, the identification model is trained based on the obtained key process variables. Thirdly, the pseudo label confidence of identification model is dynamically optimized with an historical, current, and future pseudo label confidence mean. To increase the upper limit of the identification model that is self-learning with high entropy process data, active learning is used to identify process data and diagnosis fault causes by ontology. Finally, a PCA-dynamic active safe semi-supervised support vector machine (PCA-DAS4VM) for fault identification in labeled expensive chemical processes is built. The application in the Tennessee Eastman (TE) process shows that this hybrid technology is able to: (i) eliminate chemical process noise and redundant process variables simultaneously, (ii) combine historical pseudo label confidence with future pseudo label confidence to improve the identification accuracy of abnormal working conditions, (iii) efficiently select and diagnose high entropy unlabeled process data, and (iv) fully utilize unlabeled data to enhance the identification performance.

**Keywords:** semi-supervised learning; active learning; feature selection; ontology; chemical process; fault identification

## 1. Introduction

### 1.1. Background and Significance

According to the accident statistics in chemical plants, there appear many minor anomalies before a serious accident occurs [1]. Minor abnormality generally refers to abnormal conditions, such as regulator failure or alarms caused by abnormal fluctuations. Therefore, it is of great theoretical and practical significance to conduct fault identification for chemical process to quickly discover the potential abnormality and to reliably maintain the stationary operation of chemical plants.

The existing fault identification methods are mainly divided into: Qualitative methods [2], quantitative methods [3,4], and data-driven methods [5,6]. Among all of the data-driven fault identification methods, the supervised machine learning technique provides impressive fault identification results for the chemical process [7,8]. Its fault identification accuracy rate can be as high as 92%. For example, Mohd Azlan Hussain et al. [9] proposed the kernel fisher discriminant

analysis-support vector machine (KFDA-SVM) identification model with an average accuracy of 96.79% in the Tennessee Eastman (TE) process. However, the lack of labeled process data in practice often hinders the application of supervised machine learning methods. Moreover, it is expensive to label the large amount of unlabeled process data that widely exists in chemical processes. Among them, process data represent process conditions. The labeled process data represent the process data of known process conditions, and they and their labels are used together as a training set for the identification model. For example, when a device is known to be leaking, the process data are labeled. The unlabeled process data, however, represent the process data of unknown process conditions, and only the process data are used as a training set for the identification model. For example, when it is unknown whether a device is leaking, the process data are unlabeled. How to make full use of unlabeled process data to improve the accuracy of fault identification in chemical processes is, thus, a hot topic.

Semi-supervised learning is a combination of supervised learning and unsupervised learning methods realized by the active learning strategy. It uses a small amount of labeled process data to train the initial identification model, and then improves the identification performance through a large amount of unlabeled process data. The semi-supervised learning method has been applied to various fields, such as writer identification [10], sentiment classification [11], medical image analysis [12], traffic flow [13], etc. Active learning is a sampling strategy that selects high entropy unlabeled data. First, the identification model determines the confidence of the unlabeled data. Second, the query function determines the high entropy data. Finally, unlabeled data are labeled and added to the labeled dataset. Among them, the design of the query function uses information entropy. The greater the information entropy of the data, the richer the amount of information they carry. However, the application of active learning and semi-supervised learning to chemical process fault identification is only a small presence in the literature. Wang et al. [14] proposed an active learning based semi-supervised fisher discriminant analysis (SemiFDA) model with an average fault identification accuracy of 58.10% for the TE process. Song et al. [15] proposed a dynamic spare stacked auto-encoders (DSSAE) model based semi-supervised framework. Its average fault identification accuracy is up to 90.2% for the TE process.

Some technical difficulties still exist in these methods, such as the inferiority of semi-supervised learning performances when compared to supervised ones. For this reason, Zhou et al. [16] proposed a safe semi-supervised support vector machine (S4VM). Bernhard Sick et al. [17] showed that a semi-supervised support vector machine (SemiSVM) can well exploit structure information in data and greatly improve identification performance with unlabeled data. On the other hand, the large number of instruments in the actual chemical process brings noise and redundant process variables. Therefore, this paper selects the key process variables based on PCA. In order to further improve the upper limit of S4VM self-learning, high entropy process data are selected based on active learning to realize process data fault diagnosis.

### 1.2. Method

This paper proposes a principal component analysis and dynamic active safe semi-supervised support vector machines (PCA-DAS4VM) based fault identification method. Firstly, PCA [18,19] determines the key process variables with a sum weight of more than 80%. The selection of the weight threshold is based on experience with no specific standard in practice. Secondly, the pseudo label confidence of the DAS4VM model is dynamically optimized based on the pseudo label confidence mean of concerned process data. Then, high entropy unlabeled data are selected, and an ontology-based graphical scenarios object model is used for fault diagnosis of unlabeled process data. Finally, the hybrid technology is applied to the fault identification of the TE process.

### 1.3. Contributions

The main contributions of this work are as follows. After S4VM and active learning are integrated, the pseudo label confidence of active learning is dynamically optimized, and thus, the selection accuracy of the high entropy process data is improved. By making full use of the process implicit information

and expert knowledge, the graphical scenario object model helps to determine process data labels. The root cause of the failure is analyzed through the influence relationship between variables based on process data, thereby determining the process conditions. Compared with supervised learning using the same amount of labeled data, the method fully utilizes the structure of unlabeled data to improve the accuracy of the identification model, and it ensures that the performance does not decrease when used to label expensive chemical processes.

### 1.4. Organization

The rest of the paper is organized as follows. The following part introduces the PCA-DAS4VM based fault identification framework and its implementation process. The case study part describes its practical application in the TE process. The last section shows the main conclusions and future improvement of the research.

## 2. Methods

### 2.1. Fault Identification Framework

The PCA-DAS4VM identification framework is shown in Figure 1. The first step is offline training:

(1)   Historical process data are acquired and preprocessed, including process data identification, training sets labeling, etc.
(2)   Based on PCA, feature selection of offline data is performed to extract the largest linear independent variable group **M**.
(3)   Datasets in **M** are divided into labeled dataset and unlabeled dataset.
(4)   The labeled data and unlabeled data in **M** are used to train DAS4VM.
(5)   The PCA-DAS4VM model will be built if the pseudo label confidence is higher than 80%. Otherwise, unlabeled data with high entropy are selected by active learning, and then the fault cause for unlabeled data is determined by the graphical scenario object model and added to the label dataset.
(6)   The optimal parameters of the PCA-DAS4VM identification model are lastly saved.

The second step is online fault identification. Based on the optimal PCA-DAS4VM model, the key variable data in the **M** group are acquired online to determine whether the system is running normally.

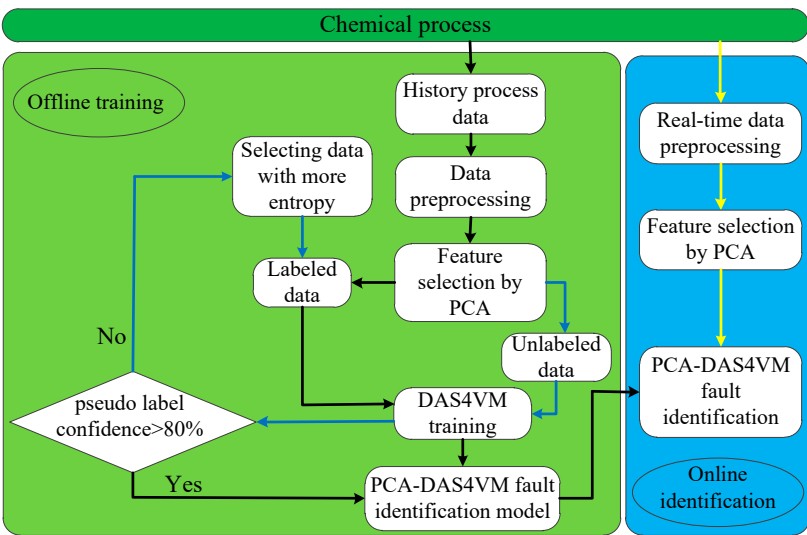

**Figure 1.** The framework of the PCA-dynamic active safe semi-supervised support vector machine (PCA-DAS4VM) based fault identification method.

### 2.2. Principal Component Analysis

Assume that there is an *m*-dimensional variable vector in the data matrix $(x_{1,g}, x_{2,g} \ldots x_{m,g})$, and each variable has *n* data $(x_{q,1}, x_{q,2} \ldots x_{q,n})^{\mathrm{T}}$, where $q = 1, 2, \ldots, m$; $g = 1, 2, \ldots, n$. For dimensionality reduction of the data matrix, PCA is used to convert the correlated variables in the data matrix to a set of linearly independent variable group **M**. The calculation steps are as follows:

(1)  Normalize Z-score of the data matrix, as shown in Equation (1).

$$Z_{qg} = \frac{x_{qg} - \mu}{\sigma} \tag{1}$$

where $\mu$ is the mean of the data matrix, and $\sigma$ is the variance of the data matrix.

(2)  Calculate the feature covariance matrix of the data matrix and then its eigenvalues, eigenvectors, and variance contribution rate, as shown in Equations (2)–(4). The variance contribution rates are sorted from large to small so that the variables whose sum of variance contributions exceeds the set ratio threshold are set as the principal component variables.

$$\mathbf{C} = \frac{1}{m} Z^T Z \tag{2}$$

$$|\mathbf{C} - \lambda \mathbf{E}| = 0 \tag{3}$$

$$\alpha_d = \lambda_d \Big/ \sum_d^o \lambda_d \tag{4}$$

where **C** is the calculated feature covariance matrix, $\lambda$ is the eigenvalue, **E** is the identity matrix, *o* is the number of the principal component, and $\alpha$ is the variance contribution rate.

(3)  The linear expression of the principal component is established, and the coefficients of variables in the linear expression of each principal component are determined, as shown in Equation (5).

$$coe_{q,d} = \frac{v_{q,d}}{\sqrt{e_d}} \tag{5}$$

where $coe_{q,d}$ is the coefficient of the *q*th variable in the linear expression of the *d*th principal element, $v_{q,d}$ is the *d*th principal component of the *q*th variable, and $e_d$ is the eigenvalue of the *d*th principal component.

(4)  Obtain a comprehensive score model based on the coefficients of principal component variables in the principal element linear expression, as shown in Equation (6).

$$w_q = \sum_{d=1}^{o} coe_{q,d} \times s_d \Big/ \sum_{d=1}^{o} s_d \tag{6}$$

where $w_q$ is the coefficient of the *q*th variable in the comprehensive scoring model, and $s_d$ is the variance of the *d*th principal component.

(5)  Normalize the variable coefficient in the comprehensive score model and redefine the weight of the variables.

### 2.3. Dynamic Active Safe Semi-Supervised Support Vector Machine

S4VM is a safe semi-supervised learning algorithm that can improve the identification accuracy by using a large amount of easily accessible unlabeled data. The term "safe" here means, in theory, that S4VM is not inferior to the supervised learning algorithm that uses the same amount of labeled data. In S4VM, it is believed that there are often multiple similar low-density maximum interval classifiers in a data space (see Figure 2), any one of which may become the optimal classifier later.

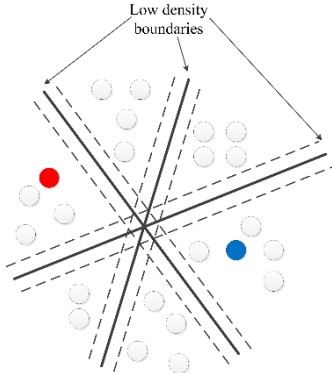

**Figure 2.** S4VM fault identification diagram focusing on multiple low density boundaries.

The traditional semi-supervised SVM algorithm attempts to obtain an optimal solution by using a decision function $f$ assigned to an unlabeled dataset. The following loss function (Equation (7)) is used, and a maximum interval classifier can be obtained by solving Equation (7).

$$h(f, \hat{y}) = \min_{f, \hat{y}} \frac{1}{2}\|f\|^2 + C_1 \sum_{i=1}^{l} l(y_i, f(x_i)) + C_2 \sum_{j=i+1}^{l+u} l(\hat{y}_j, f(x_j)) \tag{7}$$

where $x$ is the input space, $y$ is the labeled space, $\hat{y}$ is the prediction space, $i$ is the labeled data ($i$ = 1, 2, $\ldots$ , $l$), $j$ is the unlabeled data ($j$ = 1, 2, $\ldots$ , $u$), and $C_1$ and $C_2$ are hyperparameters with labeled data and unlabeled data, respectively.

In order to obtain a set containing multiple low-density maximum interval classifiers, the loss function is redefined as Equation (8).

$$\min_{\{\mathbf{w}_t, b_t, \hat{y}_t \in\}_{t=1}^T} \sum_{t=1}^{T} (\tfrac{1}{2}\|\mathbf{w}_t\|^2 + C_1 \sum_{i=1}^{l} \xi_i + C_2 \sum_{j=l+1}^{l+u} \xi_j) + M \sum_{1 \leq t \neq \bar{t} \leq T} I(\tfrac{\hat{y}_t'\hat{y}_{\bar{t}}}{u} \geq 1 - \varsigma)$$
$$s.t. \quad y_i(w'_t \phi(x_i) + b_t) \geq 1 - \xi_i, \quad \xi_i \geq 0,$$
$$\hat{y}_{t,j+l}(w'_t \phi(x_{j+l}) + b_t) \geq 1 - \xi_{j+l}, \quad \xi_{j+l} \geq 0. \tag{8}$$

where $\mathbf{W}_t$ is one dimensional vector, $b_t$ is the bias, $t$ is the classifier ($t$ = 1, 2, $\ldots$ , $T$), $\zeta$ is the relaxation vector, $M$ is a constant, $I$ is the indication function, and $\varsigma = 0.5$, $\phi(\,)$ is a feature mapping guided by a kernel function.

According to the smoothness assumption of semi-supervised learning, the historical pseudo label confidence $p_{j-1}$ and the future pseudo label confidence $p_{j+1}$ constitute dynamic related information to reduce misjudgment of the current working condition, as shown in Equation (9).

$$p_{j,k} = \frac{1}{3}(p_{j-1,k} + p_{j,k} + p_{j+1,k}) \tag{9}$$

where the confidence that the $j$th data belong to the $k$th class is $p_{jk}$.

In general, semi-supervised learning is based on a small number of labeled data. It can improve performance by utilizing useful data distribution information provided by a large amount of unlabeled data, but it may also lead to misleading or even wrong information due to data noise. Therefore, the unlabeled data are selected according to their entropy. The higher the entropy is, the greater the amount of information is carried by the process data, in which the entropy of unlabeled data $j$ is calculated by Equation (10).

$$ent_j = -\sum_{k=1}^{K} p_{jk} \log p_{jk} \tag{10}$$

where $ent_j$ is the $j$th data entropy, and $k$ is the number of classes ($k = 1, 2, \ldots, K$).

The DAS4VM selects the unlabeled data with high entropy, where the criteria for selection is given in Equation (11).

$$x_s = \arg\max_{x_j \in \mathbf{X}_U} ent_j \tag{11}$$

The criterion for DAS4VM to stop selecting unlabeled data is given in Equation (12).

$$\max_{1 \leq k \leq K} p_{j,k} \geq a \tag{12}$$

where $a$ is the stopping threshold.

The pseudo code of DAS4VM is shown in Algorithm 1.

---

**Algorithm 1** DAS4VM

---

**Input:** $D = \{\{x_i, y_i\}_{i=1}^{l}, \{x_j\}_{j=l+1}^{l+u}\}$;

**Output:** y.

1: Optimize pseudo label confidence through dynamic related information via Equation (9).

2: Calculate the entropy of unlabeled data and sort them via Equations (10)–(12).

3: Determine the fault cause of the selected unlabeled data by graphical scenario object model (see Section 2.4).

4: Generate a pool of diverse large-margin low-density separators $\{\hat{y}\}_{t=1}^{T}$ for D via Equation (8).

5: Assign the labels $y = \{y_{l+1}, \ldots, y_{l+u}\}$ to unlabeled instances such that improvement in performance for any separator $\hat{y}_t$ ($t = 1, \ldots, T$) is maximized.

---

### 2.4. Ontology

The graphical scenario object model based on ontology determines the failure reason of the unlabeled process data. The ontology [20] is a clear norm of conceptual expression, which mainly includes the types of structural information, basic elements, and expressions of influence relations. It can systematically and graphically express the chemical process, facilitating domain knowledge sharing between people and computers. When an anomaly occurs in the chemical process due to the process parameter's deviation, the anomaly will propagate along the material flow, energy flow, and information flow under the deviation. If no preventive action is made in time, the accident will likely occur. The process described above is a hazard scenario [21].

## 3. Case Study

### 3.1. Process Description

The proposed method is applied to the TE benchmark process. The TE process was given by Downs and Vogel from Tennessee Eastman Chemical Company [22]. The TE process consists of five main operating units: Reactor, condenser, compressor, separator, and stripper, as shown in Figure 3. The rectangles in Figure 3 represent measuring instruments, such as PI03 for the pressure of the stripper, while the circles in Figure 3 represent control instruments, such as FIC01 for the flowrate of stream 1. The heat and mass balance of the TE process is shown in Table 1. The process includes 22 continuous measured variables (Table 2) and 20 preset fault modes (Table 3).

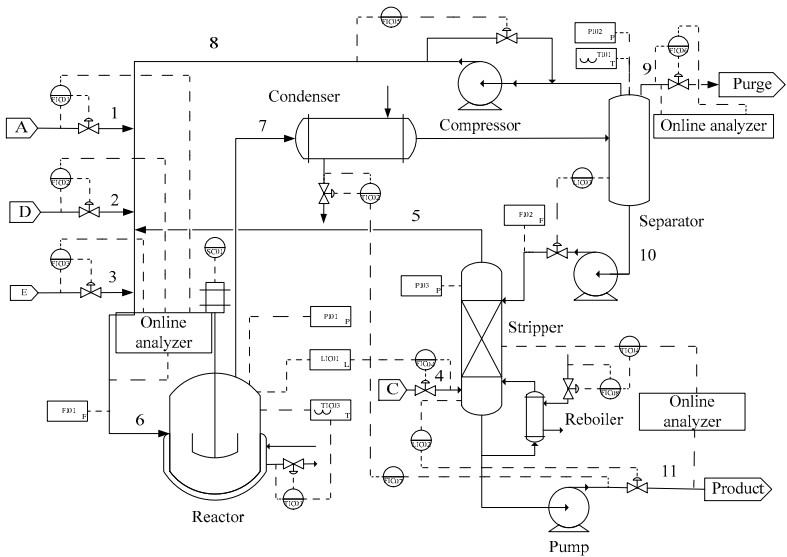

**Figure 3.** Flow chart of Tennessee Eastman (TE) process.

**Table 1.** TE process heat and mass balance table.

| No. in Process | Flow /koml·h$^{-1}$ | Temperature/°C | Process Concentration/mol·mol$^{-1}$ | | | | | | | |
|---|---|---|---|---|---|---|---|---|---|---|
| | | | **A** | **B** | **C** | **D** | **E** | **F** | **G** | **H** |
| 1 | 11.2 | 45.0 | 0.99990 | 0.00010 | 0.00000 | 0.00000 | 0.00000 | 0.00000 | 0.00000 | 0.00000 |
| 2 | 114.5 | 45.0 | 0.00000 | 0.00010 | 0.00000 | 0.99990 | 0.00000 | 0.00000 | 0.00000 | 0.00000 |
| 3 | 98.0 | 45.0 | 0.00000 | 0.00000 | 0.00000 | 0.00000 | 0.99990 | 0.00010 | 0.00000 | 0.00000 |
| 4 | 417.5 | 45.0 | 0.48500 | 0.00500 | 0.51000 | 0.00000 | 0.00000 | 0.00000 | 0.00000 | 0.00000 |
| 5 | 465.7 | 65.7 | 0.43263 | 0.00444 | 0.45264 | 0.00116 | 0.07256 | 0.00885 | 0.01964 | 0.00808 |
| 6 | 1890.8 | 86.1 | 0.32188 | 0.08893 | 0.26383 | 0.06882 | 0.18776 | 0.01657 | 0.03561 | 0.01659 |
| 7 | 1476.0 | 1476.0 | 0.27164 | 0.11393 | 0.19763 | 0.01075 | 0.17722 | 0.02159 | 0.12302 | 0.08423 |
| 8 | 1201.5 | 1201.5 | 0.32958 | 0.13823 | 0.23978 | 0.01257 | 0.18579 | 0.02263 | 0.04844 | 0.02299 |
| 9 | 15.1 | 15.1 | 0.32958 | 0.13823 | 0.23978 | 0.01257 | 0.18579 | 0.02263 | 0.04844 | 0.02299 |
| 10 | 259.5 | 259.5 | 0.00000 | 0.00000 | 0.00000 | 0.00222 | 0.13704 | 0.01669 | 0.047269 | 0.37136 |
| 11 | 211.3 | 211.3 | 0.00479 | 0.00009 | 0.01008 | 0.00018 | 0.00836 | 0.00099 | 0.53724 | 0.43828 |

**Table 2.** Continuous measurement variable (CMV) of the TE process.

| Variable | Description | Variable | Description |
|---|---|---|---|
| CMV (1) | A Feed | CMV (12) | Separator level |
| CMV (2) | D Feed | CMV (13) | Separator pressure |
| CMV (3) | E Feed | CMV (14) | Separator underflow |
| CMV (4) | A and C Feed | CMV (15) | Stripper level |
| CMV (5) | Recycle flow | CMV (16) | Stripper pressure |
| CMV (6) | Reactor feed | CMV (17) | Stripper underflow |
| CMV (7) | Reactor pressure | CMV (18) | Stripper temperature |
| CMV (8) | Reactor level | CMV (19) | Stripper steam flow |
| CMV (9) | Reactor temperature | CMV (20) | Compressor work |
| CMV (10) | Purge flow | CMV (21) | Reactor cooling water outlet temperature |
| CMV (11) | Separator temperature | CMV (22) | Condenser cooling water outlet temperature |

**Table 3.** Preset faults of the TE process.

| No. in TE | Fault Reason | Fault Type |
|:---:|:---:|:---:|
| 1 | A/C feed ratio, B composition constant | Step |
| 2 | B composition, A/C ratio constant | Step |
| 3 | D feed temperature | Step |
| 4 | Reactor cooling water inlet temperature | Step |
| 5 | Condenser cooling water inlet temperature | Step |
| 6 | A feed loss | Step |
| 7 | C header pressure loss-reduced availability | Random |
| 8 | A, B, and C feed composition | Random |
| 9 | D feed temperature | Random |
| 10 | C feed temperature | Random |
| 11 | Reactor cooling water inlet temperature | Random |
| 12 | Condenser cooling water inlet temperature | Random |
| 13 | Reactor kinetics | Drift |
| 14 | Reactor cooling water valve | Viscous |
| 15 | Condenser cooling water valve | Viscous |
| 16 | Unknown | Unknown |
| 17 | Unknown | Unknown |
| 18 | Unknown | Unknown |
| 19 | Unknown | Unknown |
| 20 | Unknown | Unknown |

### 3.2. Feature Selection by PCA

Take fault 4 as an example. According to Equations (2)–(4), the variances and eigenvalues of the principal components in fault 4 are obtained and shown in Figures 4 and 5.

Figure 4 shows that the total variance contribution rate of the first 12 principal components has reached 83.15% (more than 80%), so the first 12 principal components can reflect information about all variables. Figure 5 shows the eigenvalues of the first 12 principal components as well.

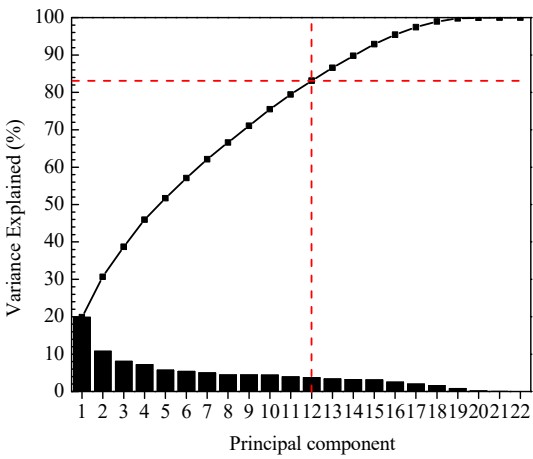

**Figure 4.** Principal component variance percent explained for fault 4.

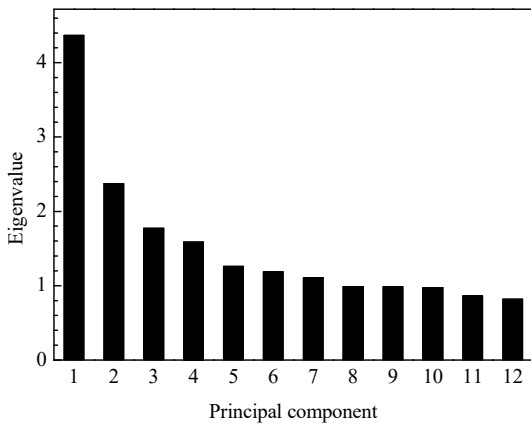

**Figure 5.** The principal component eigenvalues of fault 4.

Then, according to Equations (5) and (6), the comprehensive score coefficient and weight ratio of all the variables are obtained based on the determined 12 principal components in fault 4. The sum of the weights of the first 13 variables of fault 4 is 80.06% (more than 80%), so the first 13 variables represent all variables, as shown in Figure 6. Key variables for other faults are similarly selected when the sum of their weights are more than 80%, as shown in Figure 7.

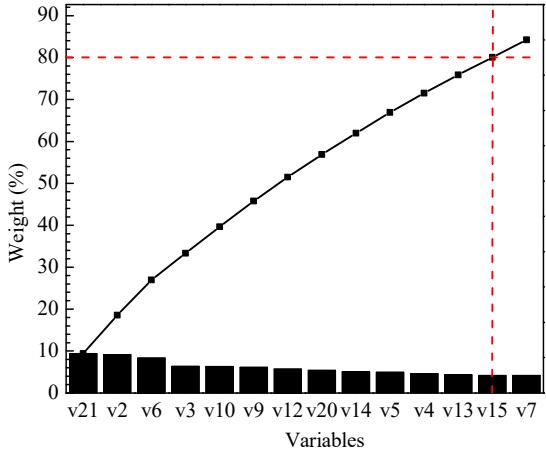

**Figure 6.** Weight of the key measurement variables for fault 4.

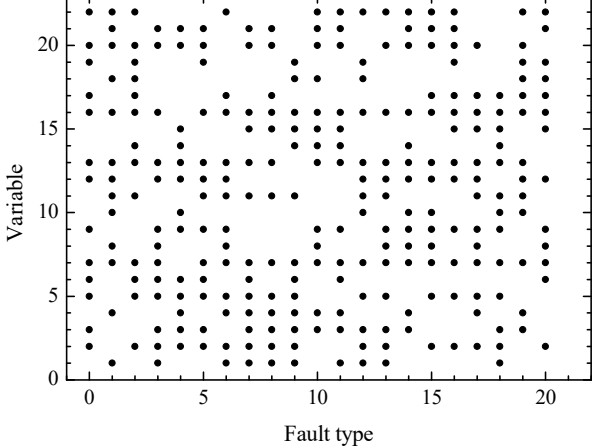

**Figure 7.** Key variables selected for each of the 20 TE faults.

### 3.3. Identification Results and Discussion

In recent years, many researchers have made various optimization efforts for supervising SVM and have achieved remarkable results. For example, Gao et al. [23] used grid search to determine the optimal SVM parameters. Yuan et al. [24] optimized SVM parameters with the cuckoo algorithm. Xiao [25] trained KNN and SVM on the same training set, and then used the integrated model of KNN and SVM to predict the test dataset. KNN predict the test dataset through labels of k nearest neighbors data, SVM predict the test dataset based on hyperplane. However, these methods are still limited to supervised learning, resulting in poor generalization performance, a low industrial fault diagnosis rate (FDR), and a high false positive rate (FPR). Therefore, this paper proposes a fault identification method PCA-DAS4VM based on a graphical scenario object model, which improves the identification accuracy of traditional SVM due to its full use of the unlabeled data distribution information. In order to better prove the effectiveness of the proposed method, the PCA-DAS4VM proposed in this paper is compared with the DAS4VM and PCA-S4VM fault identification methods when applied to the TE process. In these methods, the DAS4VM directly recognizes raw process data, and the PCA-S4VM model is based on S4VM to identify key process data selected by PCA.

In order to clearly show the performance of the proposed method, this paper defines a confusion matrix (Table 4), F1 score (Equation (13)), fault diagnosis rate (FDR) (Equation (14)), false positive rate (FPR) (Equation (15)), and accuracy (Equation (16)) as comparing criterions.

$$F1 = \frac{2 \times PRE \times REC}{PRE + REC} \times 100\% \tag{13}$$

$$FDR = \frac{TP}{TP + FN} \times 100\% \tag{14}$$

$$FPR = 1 - \frac{TN}{FP + TN} \times 100\% \tag{15}$$

$$Accuracy = \frac{TN + TP}{FP + TN + TP + FN} \times 100\% \tag{16}$$

where TN represents a normal condition diagnosed as normal, FP represents a fault condition diagnosed as normal, FN represents normal conditions diagnosed as a fault, and TP represents a fault condition diagnosed as a fault.

**Table 4.** Confusion matrix.

|  | Normal Condition | Failure Condition |
| --- | --- | --- |
| Diagnosed as normal | TN | FP |
| Diagnosed as failure | FN | TP |

Where TN represents a normal condition diagnosed as normal, FP represents a fault condition diagnosed as normal, FN represents normal conditions diagnosed as a fault, and TP represents a fault condition diagnosed as a fault.

Semi-supervised learning is usually sensitive to the number of labeled data and unlabeled data. This paper uses the S4VM parameters recommended by Li et al. [26], as shown in Table 5. The parameter "a" is recommended by Yin et al. [14]. The amount of labeled data and unlabeled data are determined based on experience and experiments. Since the amount of labeled data and unlabeled data directly affect the accuracy of the identification model, the influence of the number of labeled data and unlabeled data on the identification accuracy is discussed with the average identification accuracy of 20 kinds of TE faults as criterion, as shown in Figures 8 and 9.

First, under the condition that the training set contains 960 unlabeled data, the impact of the amount of labeled data in the training set on the accuracy of the identification model is tested. It can be seen from Figure 8 that with the increasing number of labeled data, the average identification accuracy of PCA-DAS4VM is gradually improved. However, the number of labeled data increases slowly after more than 10, so this article uses 10 labeled data (especially denoted by red color in Figure 8). Secondly,

under the condition that the training set contains 10 labeled data, the impact of the number of unlabeled data in the training set on the accuracy of the identification model is tested. All of the number of unlabeled data for each condition is 960. It can be seen from Figure 9 that with the increasing percentage of unlabeled data, the average identification accuracy of PCA-DAS4VM is gradually improved, so this paper uses the 100% (960) unlabeled data to train PCA-DAS4VM (especially denoted by red color in Figure 9). Therefore, the training set of PCA-DAS4VM contains 10 labeled data and 960 unlabeled data.

**Table 5.** Parameters of PCA-DAS4VM.

| Title 1 | Title 2 |
|---|---|
| Sample Time | 100 |
| $C_1$ | 100 |
| $C_2$ | 0.1 |
| kernel | RBF |
| a | 0.8 |

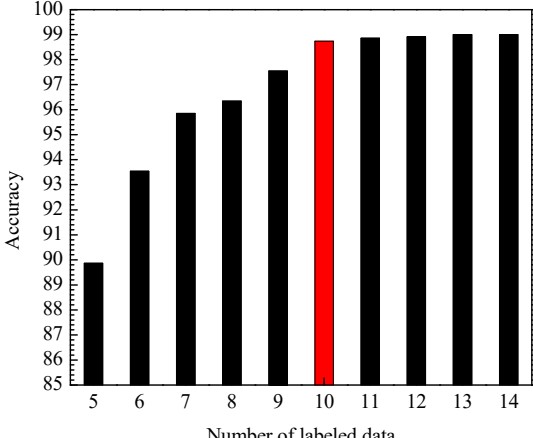

**Figure 8.** Accuracy of PCA-DAS4VM under different numbers of labeled data.

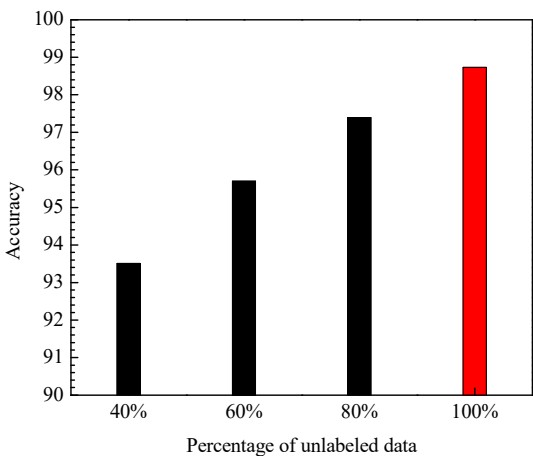

**Figure 9.** Accuracy of PCA-DAS4VM under different percentages of unlabeled data.

The F1 scores for PCA-S4VM, DAS4VM, and PCA-DAS4VM using 20 types of faults in the TE process are compared in Figure 10. It can be seen from Figure 10 that compared with other identification methods, PCA-DAS4VM has the highest F1 score. Compared with PCA-S4VM and DAS4VM, the average F1 scores of PCA-DAS4VM are enhanced by approximately 6.01% and 3.21%, respectively. The identification performance of PCA-DAS4VM is stable around 98%, further proving the reliability and stability of the S4VM. For the four types of faults in the TE process, PCA-S4VM

and PCA-DAS4VM have higher identification accuracy for drift type faults, with average F1 scores of 92.96% and 99.25%. DAS4VM has higher identification accuracy for step type faults, with an average F1 score of 95.61%.

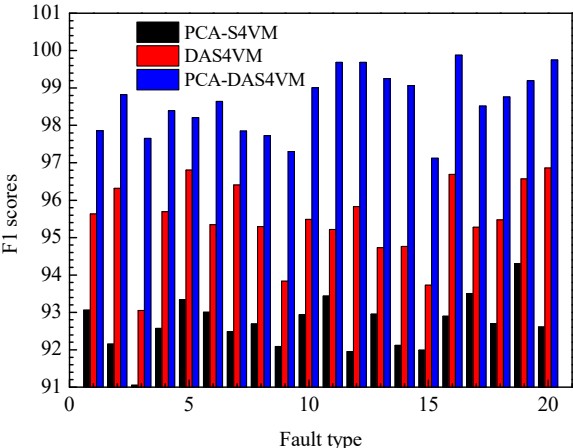

**Figure 10.** F1 scores for PCA-S4VM, DAS4VM, and PCA-DAS4VM models.

To further illustrate the effectiveness of PCA-DAS4VM for industrial processes, the FPR for the model PCA-S4VM, DAS4VM, and PCA-DAS4VM are compared in Figure 11. In Figure 11, the average FPR for PCA-DAS4VM is only 0.42%, 507.14% less than that of PCA-S4VM and 261.90% less than that of DAS4VM. For the four types of faults in the TE process, PCA-S4VM and DAS4VM have lower FPRs for step type faults, with an average FPR of 2.44% and 1.35%. PCA-DAS4VM has a lower FPR for drift type faults, with an average FPR of 0.31%.

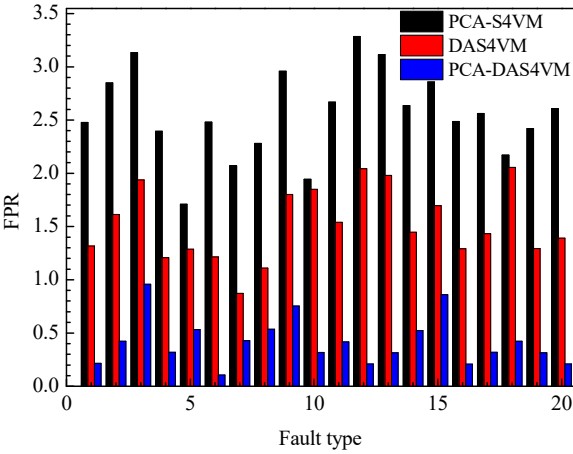

**Figure 11.** False positive rate (FPR) for PCA-S4VM, DAS4VM, and PCA-DAS4VM models.

The FDR for PCA-S4VM, DAS4VM, and PCA-DAS4VM are compared in Figure 12. The average FDR for PCA-DAS4VM is 9.35% higher than that of PCA-S4VM and 5.05% than that of DAS4VM. For the four types of faults in the TE process, PCA-S4VM and PCA-DAS4VM have higher FDRs for drift type faults, with an average FDR of 89.66% and 98.88%. DAS4VM has a higher FDR for step type faults, with an average FDR of 92.93%.

The core work of this paper is to train and test the identification models in the offline phase. In order to compare the performance of each identification model more clearly, this article compares the average computation time of each identification model for the 20 TE process faults. As can be seen from Table 6, the average computation time of PCA-DAS4VM is 130.35 s (tested on a Core i5, 8 GB memory computer) because it has high modeling complexity. PCA-DAS4VM reduces the computing

time by 53.53% compared with DAS4VM. Although PCA-DAS4VM adds 17.95% more computing time than PCA-S4VM, PCA-DAS4VM is still recommended as it has better identification performance.

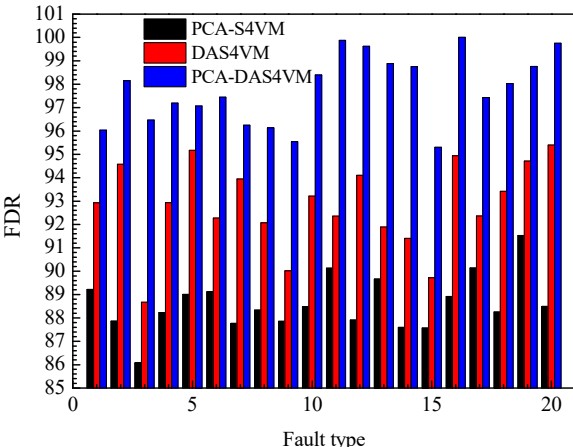

**Figure 12.** FDR for PCA-S4VM, DAS4VM, and PCA-DAS4VM models.

**Table 6.** Average computation time of PCA-S4VM, DAS4VM, and PCA-DAS4VM models.

| Method | Computation Time (s) |
|---|---|
| PCA-S4VM | 110.51 |
| DAS4VM | 280.52 |
| PCA-DAS4VM | 130.35 |

In order to better demonstrate the effectiveness of the PCA-DAS4VM method, this method is also compared with those typical semi-supervised learning methods such as ALSemiFDA [14] and DSSAE [15]. The accuracy rate of each method when applied to TE process is shown in Figure 13. It can be concluded that the accuracy rate of the PCA-DAS4VM method is 33.76% and 6.74% higher than that of ALSemiFDA and DSSAE, respectively.

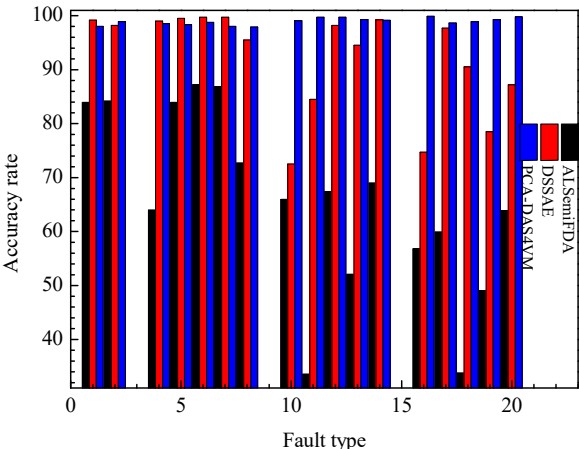

**Figure 13.** Accuracy rate for dynamic sparse stacked auto-encoders (DSSAE), ALSemiFDA, and PCA-DAS4VM models.

For fault diagnosis, the TE process is divided into five parts for analysis: Reactor, condenser, separator, recycle compressor, and stripper. The graphical scenarios object model of TE process based on ontology is established (Figure 14) where the circle point represents 22 continuous measurement variables (Table 2) of the TE process, and the connection lines indicate mutual influence between

variables (also called events). The mutual influence includes four relations: Control relation, reaction relation, type relation, and position relation [27].

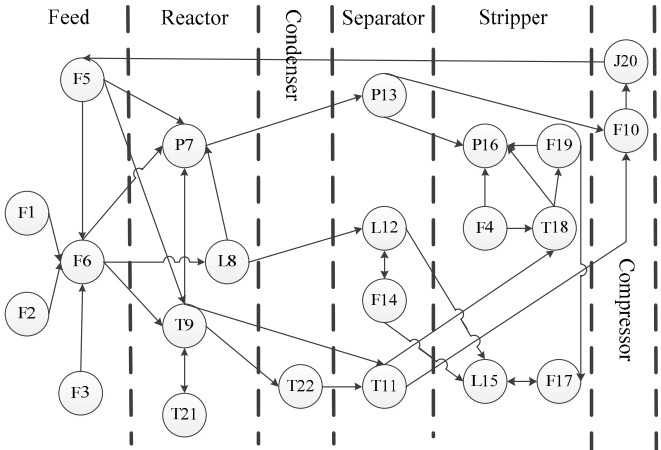

**Figure 14.** Graphical scenario object model of the TE process based on ontology.

This scenario object model is used to monitor the continuous measurement variables in the TE process, starting with the key events of the initial alarm, and then reversely reasoning the possible root nodes. Take fault 4 as an example. The reactor temperature in fault 4 is the first variable to have a high alarm. As the reaction is an exothermic reaction, three direct causes possibly affecting the reactor temperature are found: Recycle flow abnormality, reactor feed rate abnormality, and reactor cooling water temperature abnormality. This fault leads to four direct consequences: High reactor cooling water outlet temperature, high condenser cooling water outlet temperature, high product separator temperature, and high reactor pressure. The graphical scenario object model of fault 4 is shown in Figure 15.

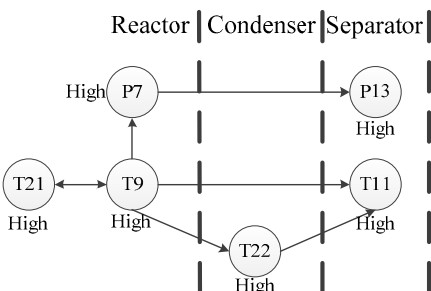

**Figure 15.** Fault 4 graphical scenario object model.

After analysis, we find that the circulating flow rate and the reactor feed flow rate have no deviation, so the abnormal temperature of the reactor cooling water most likely causes the reactor temperature anomaly. Therefore, the root cause of fault 4 is the abnormal temperature of the reactor cooling water.

## 4. Conclusions

The powerful capability of PCA in feature selection, DAS4VM in fault identification, and the graphical scenario object model in fault diagnosis are combined in this paper. Under the premise that the process data are complete and all the instruments are normal, the method can accurately identify the failure of the industrial process and alert the operator as soon as possible for the failure. The graphical scenario object model established by this method helps operators understand the process

failure propagation mechanism. This work aims for full use of the unlabeled data structure while it is in the process of improving the generalization performance and identification accuracy of fault identification. TE process data are non-stationary data but our model is fit for non-stationary data, although it does not specifically deal with non-stationary data. The superiority of the method is demonstrated by comparison with traditional semi-supervised learning methods such as DSSAE and ALSemiFDA. The average accuracy of this method is 98.93%, which is 33.76% and 6.74% higher than ALSemiFDA and DSSAE, respectively. Its average F1 scores are 6.01% and 3.21% greater than that of PCA-S4VM and DAS4VM, respectively.

How to improve the adaptability and anti-interference of the identification method is the future focus of this research.

**Author Contributions:** Methodology, X.J.; Project Administration, Z.L. and H.W.; Writing—Original Draft Preparation, X.J.; Writing—Review and Editing, X.J., W.T., X.Y. and C.L.; Validation, X.J., W.T., X.Y. and C.L. All authors have read and agreed to the published version of the manuscript.

**Funding:** Financial support for carrying out this work was provided by the National Natural Science Foundation of China (Grant No. 21576143 and 21706291), Key Research and Development Program of Shandong Province (Grant No. 2018YFJH0802).

**Conflicts of Interest:** The authors declare no conflict of interest.

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
