# Peer review of "A Dynamic Active Safe Semi-Supervised Learning Framework for Fault Identification in Labeled Expensive Chemical Processes"

_processes, doi:10.3390/pr8010105_

Round 1
Reviewer 1 Report
The manuscript is an interesting one. However, it should be improved before publication. My comments are the following:
Do you mean alarms? Or you should describe what minor anomalies mean.
Page 1 The authors said "there appear many minor anomalies" What do you mean by that? Do you mean alarms? Page 2 What do you mean about unlabelled process data. I think all the process information should have some tag (for example P101 loop OP) which will make it labelled. Page 2 What is semi-supervised learning method? The authors tell examples of concepts without describing them. This is the main problem through all the manuscript. Every concept should be described at first use, or the manuscript becomes harder to read and understand. Page 2 In the methodology section, the authors state: "Firstly, PCA [18,19] 60 determines key process variables with a sum of weight more than 80%..." How was this value chosen? Page 6 The process description should be more detailed, including the measurement points and a readable PFD. Page 7 All the references should be addressed with at least a half-sentence, lumped references should be avoided. Page 8 What is the F1 score? This is the first occurrence and also not described. Page 8-9 If the variables are not continuous, there should not be a continuous line between them on figures. Maybe the authors should use bigger markers. Page 9 The FDR and FPR should be described, and the authors should show in every case which value is better and why. Page 10 Figure 13 How this plot was obtained? This should be described in details, as well as the PFD, should contain the measurement points. Page 10 The authors should discuss how the results can contribute at an industrial level. A notation should be added.In summary, the work is certainly an interesting one and worth publication. However, the readability should be improved by the detailed description of all the variables and concepts.
Reviewer 2 Report
The paper presents a semi-supervised learning scheme for fault identification in a process plant. The paper is well written and the idea is quite good but needs some minor modifications.
The distinction between labeled and unlabeled data shall be explained with examples. It needs to be clarified if the proposed approach also helps in the labeling of the unlabelled data. It requires to be explained how much prior knowledge about the process dynamics is required to apply the proposed method. In figure 8, does 100% mean 960 unlabeled data sets? Are there any guidelines for choosing the number of labeled and unlabeled data sets for better results? The computational cost of the proposed method with the existing ones needs to quantified as well. How does the proposed method work in case of non-stationary data?Author Response
Please see the attachment.

Round 2
Reviewer 1 Report
The authors answered all my questions, the manuscript can be accepted.